https://doi.org/10.1038/s41467-019-10983-7　**OPEN**

# Essentiality of sterol synthesis genes in the planctomycete bacterium *Gemmata obscuriglobus*

Elena Rivas-Marin [1,4], Sean Stettner [2,4], Ekaterina Y. Gottshall[2,4], Carlos Santana-Molina [1], Mitch Helling [3], Franco Basile[3], Naomi L. Ward[2] & Damien P. Devos [1]

Sterols and hopanoids are chemically and structurally related lipids mostly found in eukaryotic and bacterial cell membranes. Few bacterial species have been reported to produce sterols and this anomaly had originally been ascribed to lateral gene transfer (LGT) from eukaryotes. In addition, the functions of sterols in these bacteria are unknown and the functional overlap between sterols and hopanoids is still unclear. *Gemmata obscuriglobus* is a bacterium from the *Planctomycetes* phylum that synthesizes sterols, in contrast to its hopanoid-producing relatives. Here we show that sterols are essential for growth of *G. obscuriglobus*, and that sterol depletion leads to aberrant membrane structures and defects in budding cell division. This report of sterol essentiality in a prokaryotic species advances our understanding of sterol distribution and function, and provides a foundation to pursue fundamental questions in evolutionary cell biology.

[1] Centro Andaluz de Biología del Desarrollo (CABD)-CSIC, Pablo de Olavide University, Seville 41013, Spain. [2] Department of Molecular Biology, University of Wyoming, Laramie, WY 82071-2000, USA. [3] Department of Chemistry, University of Wyoming, Laramie, WY 82071-2000, USA. [4] These authors contributed equally: Elena Rivas-Marin, Sean Stettner, Ekaterina Y. Gottshall. Correspondence and requests for materials should be addressed to N.L.W. (email: nlward@uwyo.edu) or to D.P.D. (email: damienpdevos@gmail.com)

Sterols perform multiple essential functions in eukaryotic cell membranes, such as modulating membrane fluidity and permeability[1–3], and allowing the formation of liquid-ordered membrane states (lipid rafts)[4–6], which are critical for cell biology and pathogenesis. Sterol-induced membrane changes are also at the core of eukaryotic cell division, and thus, the emergence of sterols has long been recognized as a critical step in the development of the eukaryotes[7].

Hopanoids and sterols are biosynthetically and functionally related polycyclic triterpenoids[8]. Most bacteria rely on hopanoid lipids to realize functions performed by eukaryotic sterols but their functional overlap is still unclear. However, a few bacteria produce sterols[9]. The function of bacterial sterols is currently unknown and the occurrence of sterol synthesis genes in bacteria has mostly been attributed to lateral gene transfer (LGT), either from eukaryotes to bacteria, or between bacteria[9–11]. This conundrum is exacerbated by the unclear evolutionary origins of both eukaryotic and bacterial sterol synthesis, despite the former's key role in eukaryogenesis, and the importance of sterols for geochemical dating[12], where their presence is usually interpreted as evidence of eukaryotic life.

The role of sterols in bacteria is understudied. Sterol function has been investigated in the myxobacteria *Stigmatella aurantiaca*[13], which produces cycloartenol, a sterol mostly synthesized by plants. An insertion mutant of the cycloartenol synthase gene of *S. aurantiaca* was compared to the wild-type strain by transmission electron microscopy, and neither growth nor membrane structure phenotypes were detected. No mutant phenotypes were observed for other characters, such as ethanol sensitivity, swarming, or aggregation. It was thus concluded that sterol production is dispensable for growth in *S. aurantiaca*, a proteobacteria[13].

In order to broaden the studies of sterol function in bacteria, we experimentally investigated the contribution of sterol synthesis in *Gemmata obscuriglobus*, a member of the phylum *Planctomycetes*, unrelated to *Proteobacteria*. Further reasons to select *G. obscuriglobus* include that it possesses the most minimal sterol synthesis pathway yet reported, producing only lanosterol and its isomer parkeol[14]. In addition, *G. obscuriglobus* is the only planctomycete reported to produce sterols while all its relatives are potentially hopanoid producers. Also, the majority of planctomycetes have a complex endomembrane system, although it is arguably most developed in this species[15]. In eukaryotes, the membrane composition of sterols contributes to the function and organization of the cell's endomembrane system[16]. Thus, *G. obscuriglobus* is ideally suited to study the functional differences between hopanoids and sterols. Here, we address these fundamental questions through a combination of genetic, chemical, and microscopy approaches.

## Results

**Genetic interruption of sterol synthesis**. In order to determine the phenotypic effects of sterol synthesis inhibition in *G. obscuriglobus*, we first applied a genetic approach. Sterols are synthesized from squalene by the enzymes squalene monooxygenase (Sqmo) and oxidosqualene cyclase (Osc). We independently interrupted the *sqmo* and *osc* genes, which are co-located in the *G. obscuriglobus* genome[14] (Supplementary Figs. 1 and 2). Wild-type and mutant cells interrupted in non-essential genes form colonies on solid media after approximately 10 days of growth. Colonies were observed for the *sqmo* and *osc* mutants only after nearly a month. These colonies were of a more intense red color than the wild-type ones (Supplementary Fig. 3). Transconjugants were unable to grow when they were transferred to fresh solid medium or liquid broth, even in the presence of

exogenous lanosterol. Thus, the mutant cells are unable to sustain normal unlimited growth. We speculate that the sterol-depleted colonies were able to undergo a few cycles of division, although inefficiently, by dilution of residual sterols present in their membranes. In order to demonstrate that the observed phenotype was due to the lack of sterols and not to plasmid integration, a similar plasmid containing a fragment of the *sqmo* gene under its own promoter was transferred into the wild-type strain. This chromosomal integration resulted in reconstruction of the wild-type operon, alongside the disrupted version, and produced cells resembling wild-type cells (Supplementary Fig. 1). Colonies appeared following 10 days of incubation, and in contrast to the interrupted mutants, these transconjugants could be regrown in liquid and solid medium, demonstrating that the *osc* or *sqmo* gene interruption was responsible for the originally observed no-growth phenotype.

**Chemical inhibition of squalene monooxygenase**. We next applied a chemical approach that, in contrast to the genetic strategy, allowed partial inhibition of sterol synthesis, to permit further characterization of sterol depletion phenotypes. Terbinafine hydrochloride is an allylamine anti-fungal compound[17] that specifically inhibits squalene epoxidase in fungi[18]. Based on sequence homology between *G. obscuriglobus* and eukaryotic Sqmo proteins[14], we predicted that terbinafine hydrochloride could specifically inhibit the *G. obscuriglobus* Sqmo enzyme. Terbinafine treatment indeed reduced *G. obscuriglobus* growth in a dose-dependent manner (Fig. 1a), and a reduction in total sterol levels was observed (Fig. 1b), suggesting that the growth defect was specifically due to sterol biosynthesis inhibition. Supplementation of terbinafine-treated cultures with exogenous lanosterol substantially improved growth at all terbinafine doses (Fig. 1c), supporting our conclusions. Further evidence of sterol essentiality was provided by an experiment in which growth was completely suppressed by treatment with zaragozic acid, an inhibitor of squalene synthase[19], but again rescued with the provision of exogenous lanosterol (Supplementary Fig. 4). This growth rescue is consistent with that observed in the terbinafine experiment (Fig. 1c). Lipotoxicity of accumulated squalene has been reported in yeast deficient in lipid storage organelles[20], and a similar effect cannot be completely excluded for *G. obscuriglobus* at this time. However, the results of this experiment suggest that it is primarily sterol essentiality rather than squalene lipotoxicity that is responsible for the phenotypes observed in terbinafine-treated cultures and the interruption mutants.

**Electron microscopy of sterol deficient cells**. As sterols have never been reported as essential for bacterial growth, we attempted to identify phenotypic defects associated with their inhibition in *G. obscuriglobus*. Because of the lack of growth of the mutants, we investigated these defects by electron microscopy. In the *sqmo* mutant, we observed diverse striking phenotypes that have never been described in wild-type cells (Fig. 2 and Supplementary Fig. 5). A series of abnormal membrane conformations were observed, including nested vesicles, stacks of membrane layers inside the cytoplasm, multiple elongated sausage-like structures, cellular space crowded with small vesicles, or a mixture of these features (Fig. 2). We also observed numerous extracellular membranous structures, presumably cellular debris (Supplementary Fig. 5). Similar but milder phenotypes were also observed in terbinafine-treated cells (Supplementary Fig. 6). This supports the conclusion that the *sqmo* mutant phenotypes are indeed due to sterol depletion and not to the use of aged cellular material, necessitated by the slower growth of mutants. We also observed mutant cells that appear to be undergoing aberrant cell

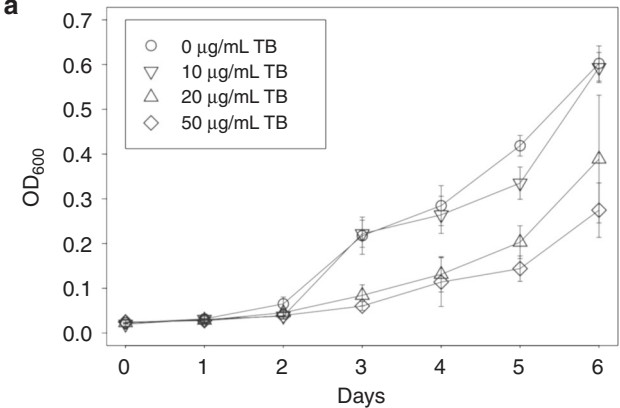

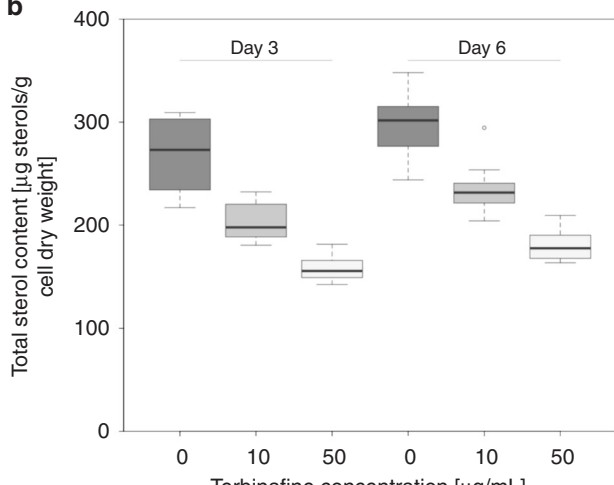

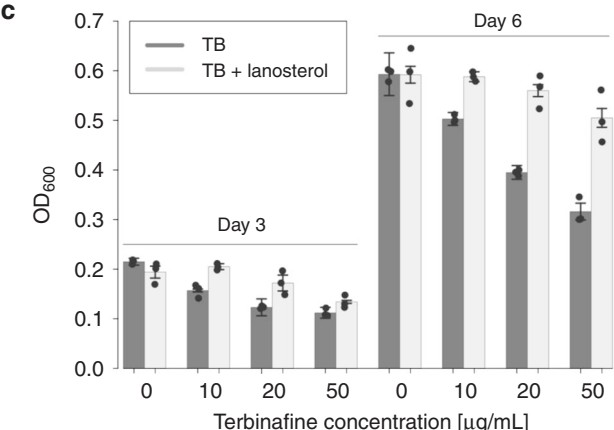

**Fig. 1** Terbinafine treatment suppresses growth and sterol production in *Gemmata obscuriglobus*, in a concentration-dependent manner. **a** Terbinafine (TB) treatment at various concentrations inhibits *G. obscuriglobus* growth relative to untreated cultures, measured spectrophotometrically over a 6-day time-course. Means and standard deviations are shown for three biological replicates. The points are represented by circles, upside-down triangles, triangles, and diamonds for increasing terbinafine concentrations, respectively. **b** Total cellular sterols, measured using gas chromatography-mass spectrometry at 3-day and 6-day culture growth time points. Data is presented in boxplot of four biological replicates, each analyzed in triplicate. The center line represents the median, the box represents the upper and lower quartiles, and the lines represent the minimum and maximum observations. Any points outside the lines are considered outliers. **c** Lanosterol supplementation of TB-treated cultures partially or completely restores growth to the level of untreated cultures, measured spectrophotometrically at 3-day and 6-day culture growth time points. Means and standard deviations are shown for three biological replicates. Source data are provided as a Source Data file

*G. obscuriglobus*[21]. The cells containing inclusions appeared to be undergoing failed budding attempts, although we cannot exclude the possibility of dynamic membrane processes other than budding. Nevertheless, normal division was not observed in any cell containing an inclusion, and some of these cells appeared to prematurely release a phase-bright bud. The presence of inclusions was also associated with external cell debris, most likely related to that observed in the *sqmo* mutant (Supplementary Fig. 5). Addition of exogenous lanosterol led to the disappearance of these structures and to the rescue of budding replication (Fig. 3c, d and Supplementary Movies 1 and 2). This rescue effect strongly suggests that sterols are required for *G. obscuriglobus* cell division, the first reported evidence for an essential role of sterols in cell division outside of the domain Eukarya. To the best of our knowledge, hopanoids have never been reported to be essential for lab growth conditions nor have they been involved in cell division in bacteria, with one possible exception[22]. Because *G. obscuriglobus* is the only planctomycete reported to produce sterols, while all its relatives are hopanoid producers, the sterol essentiality reported here opens interesting avenues to study the functional differences between these molecules.

## Discussion

We report the essentiality of bacterial sterol synthesis genes, at least under laboratory growth conditions. Our findings complement those of Gudde et al.[23], who demonstrated in *G. obscuriglobus* that chemical inhibition of the downstream sterol synthesis enzyme, oxidosqualene cyclase, resulted in cell death. Although depletion of sterol levels was not directly detected, the authors achieved rescue of the growth defect with exogenous lanosterol supplementation, supporting the sterol inhibition as causative of cell death. Our study however provides the first evidence for an association between sterol synthesis, intracellular membrane morphology, and cell division, which is also novel for bacterial species. It appears likely that future studies of *G. obscuriglobus*, and perhaps other sterol-synthesizing bacteria, will identify novel functions of sterols in the context of bacterial cell biology. This will contribute to an improved understanding of fundamental similarities and differences between prokaryotic and eukaryotic cells.

## Methods

**Bacterial strains and culture conditions**. The bacterial strains used in this work are summarized in Supplementary Table 1. *Escherichia coli* was grown in lysogeny broth (LB) medium at 37 °C, and *G. obscuriglobus* DSM5831[T] strains in LB NaCl-free medium pH 7.2 at 28 °C or PYGV medium (DSMZ 621) at 30 °C. 1.5% bacto-

division, including cells with multiple buds and closed neck between cells (Fig. 2g, h). In contrast to wild-type dividing cells, the membranes at the bud of the necks in mutant cells seem to be occluding the length of the neck (Fig. 2a, g, h).

**Time-lapse microscopy of inhibited cells**. In order to observe these cellular defects in real-time, we returned to the chemical inhibition approach, monitoring terbinafine-treated cultures through time-lapse microscopy assays (Fig. 3; Supplementary Movies 1 and 2). Terbinafine addition resulted in the development of phase-bright inclusions in approximately one-third of the cells within 4–5 h after the start of the time-lapse experiment (Fig. 3b–d), which roughly correspond to half a cell cycle in

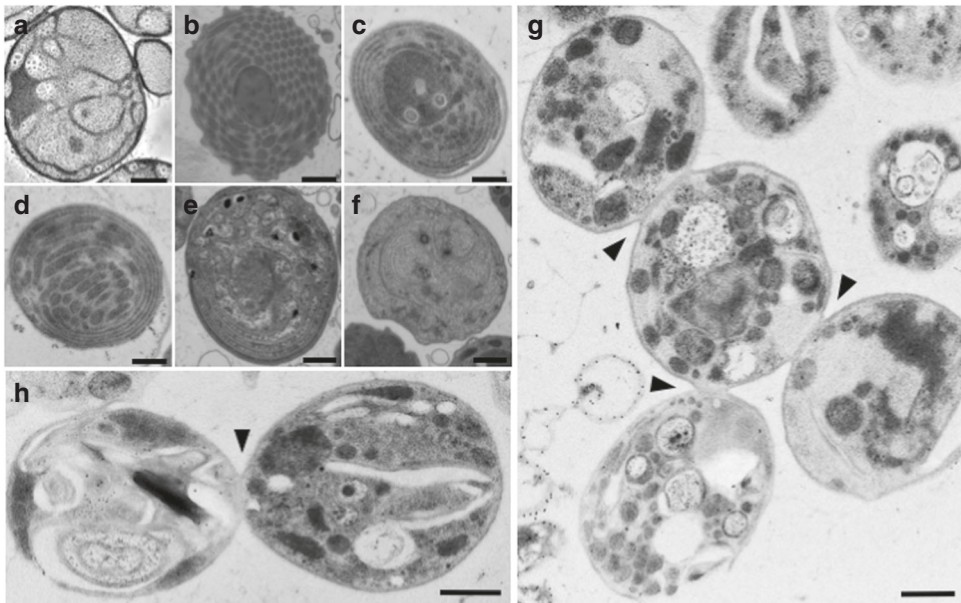

**Fig. 2** Transmission electron microscopy of *G. obscuriglobus* cells deficient in sterol synthesis. Representative membrane phenotypes observed in the *G. obscuriglobus* mutant interrupted in the *sqmo* gene. Wild-type cell (**a**). Various membrane organizations are shown, including **b** cellular space crowded with small vesicles, **c** stacks of membrane layers inside the cytoplasm, **d** multiple elongated sausage-like structures, **e**, **f** or a mixture of these and undefined features, **g** multiple budding. **g**, **h** Arrows indicate necks between dividing cells. Black dots correspond to localization of the GFP used to interrupt the *sqmo* gene. Scale bar represents 0.5 µm

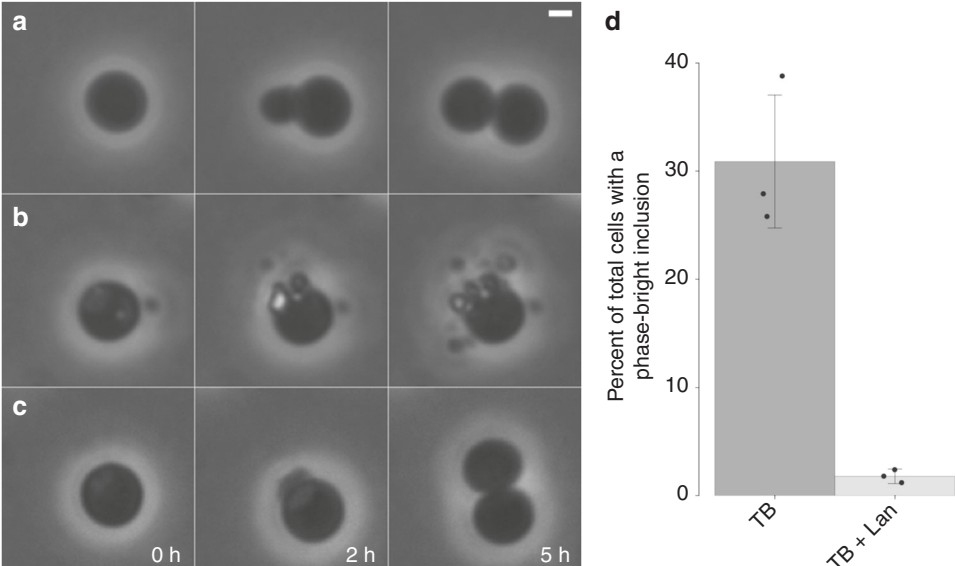

**Fig. 3** Terbinafine treatment inhibits normal budding replication in *Gemmata obscuriglobus*. Selected frames of phase-contrast time-lapse microscopy of 4-day *G. obscuriglobus* cultures. **a** Control culture. **b** Culture treated with 100 µg mL$^{-1}$ terbinafine. **c** Culture treated with 100 µg mL$^{-1}$ terbinafine and 1 µg mL$^{-1}$ lanosterol. **a**–**c** Scale bar represents 1 µm. **d** Incidence of phase-bright inclusions in TB-treated culture with or without lanosterol supplementation. Phase-bright inclusions were never observed in untreated cultures. Means and standard deviations are shown for three biological replicates. Source data are provided as a Source Data file

agar was added for solid media. To avoid contamination of *G. obscuriglobus* cultures, 50 µg mL$^{-1}$ cycloheximide (Cyclo) and 100 µg mL$^{-1}$ ampicillin (Ap) were added. *G. obscuriglobus* cultures were grown aerobically in flasks, in a volume corresponding to one-fifth of a flask. When required, kanamycin (Km) was used at the following concentrations (µg mL$^{-1}$): 30 for *G. obscuriglobus* and 25 for *E. coli*.

For determining the effects of terbinafine on the growth of *G. obscuriglobus*, an exponential-phase culture was inoculated to fresh liquid medium containing up to 50 µg mL$^{-1}$ terbinafine hydrochloride (dissolved in ethanol), or 100 µL mL$^{-1}$ ethanol (negative control) and subsequently incubated for up to 6 days. For complementation with exogenous lanosterol, PYGV medium containing up to 50 µg mL$^{-1}$ terbinafine hydrochloride and/or up to 5 µg mL$^{-1}$ lanosterol (dissolved in DMSO) was used. Optical density was measured every 24 h at wavelength of 600 nm (OD600). Each experiment contained between three and five biological replicates.

An exponential phase culture was inoculated to fresh medium at a 1:1000 dilution containing one of the following: 40 µl mL$^{-1}$ ethanol (negative control), 20 µg mL$^{-1}$ zaragozic acid (dissolved in ethanol), 5 µg mL$^{-1}$ lanosterol (dissolved in DMSO), or both lanosterol and zaragozic acid at their respective concentrations. OD600 was measured every 24 h. Each experiment contained three biological replicates.

**Plasmids**. The oligonucleotides and plasmids used in this work are summarized in Supplementary Tables 2 and 3, respectively. All DNA manipulations were made using standard protocols. Plasmid pMPO1012[24] was used as template to generate one-event homologous recombination insertion mutants. This is a ColE1-mobilizable plasmid, containing the *mut3a-gfp* gene encoding a Green Fluorescent Protein expressed under a heterologous promoter, and an *nptII* kanamycin resistance gene. In order to support homologous recombination, 1232 and 1054 bp fragments *sqmo* or *osc* genes, respectively, were amplified using genomic DNA as a template, and cloned into the pMPO1012 vector as a HindIII or HindIII/Klenow-directed fragment. The pDV011 plasmid was constructed to interrupt the *sqmo* gene by single homologous recombination using the primers Int SQMO *Gemmata* fwd and Int SQMO *Gemmata* rv (1232 bp). The pDV058 vector was built to disrupt the *osc* gene by single homologous recombination using the primers Int OSC *Gemmata* fwd and Int OSC *Gemmata* rv (1054 bp). Use of the pDV037 vector allowed interruption of the *sqmo* gene as achieved with pDV011, using the primers Int SQMO *Gemmata* 2 fwd and Int SQMO *Gemmata* rv (1511 bp), but resulting in regeneration of the full sterol synthesis operon after genome integration. The constructions are shown in Supplementary Fig. 1.

**Genetic modification: triparental mating**. Genetic transformation of *G. obscuriglobus* was performed by triparental mating using 200 μL of an exponentially growing culture (OD600 ~0.4) of the donor (Supplementary Table 2) and helper (*E. coli* DH5α pRK2013) strains in LB, with a cell pellet from 15 mL of the receptor strain culture (OD600 ~0.4). Each culture was previously individually washed in phosphate buffer and resuspended in a total volume of 100 μL of phosphate buffer that was spotted onto the corresponding agar plates containing Cyclo. The conjugation patches were incubated for 24 h at 28 °C, and resuspended in 1 mL of phosphate buffer. Trans-conjugants were plated on Ap 100 μg mL$^{-1}$ and Km 30 μg mL$^{-1}$ with or without lanosterol dissolved in chloroform (20 μg mL$^{-1}$ from a 1000× stock). Viable cells were plated on Ap 100 μg mL$^{-1}$. Colonies appeared after 10–30 days of incubation, depending on the plasmid integrated.

**Characterization of modified strains by Southern blotting**. Strains harboring the mutations was verified by fluorescence microscopy. Southern blotting analysis with 2 μg of genomic DNA was also performed for the complemented strain (DV026). Genomic DNA was extracted using the Wizard Genomic DNA Purification Kit (Promega). The digested DNA was resolved by agarose gel electrophoresis. PCR amplicon used as probe was synthetized with the primers GFP pDV020 fwd and GFP pDV020 rv (Supplementary Table 1). Probes were labeled according to the manufacturer's instructions (DIG DNA Labeling Kit, Roche). DIG-labeled probes were detected with anti-Digoxigenin-AP, Fab fragments and CSPD (Roche). Visualizations were performed using Chemidoc XRS, and images were analyzed with the ImageLab 5.0 software.

**Sterol extraction and derivatization**. *G. obscuriglobus* pre-grown for 4 days in PYGV medium was used to inoculate 300 mL of fresh medium, supplemented with final concentrations of up to 50 μg mL$^{-1}$ terbinafine and/or up to 1 μg mL$^{-1}$ lanosterol. All cultures contained Ap and Cyclo to a final concentration of 100 μg mL$^{-1}$ and 50 μg mL$^{-1}$, respectively. Two separate sets of cultures were grown for incubation periods of 3 days and 6 days. After incubation, pelleted cells were washed three times with phosphate-buffered saline, one time with EtOH to remove residual terbinafine, and finally one time with ddH$_2$O to remove salts. All washes were performed with reagents at 4 °C. Pellets were flash frozen in liquid nitrogen and lyophilized for 24 h (Freezone 6 Freeze Dry System Labconco). Dried pellets were kept at −20 °C.

Dried pellets were weighed into 10 mL glass vessels (CEM Corp.) to obtain the dry weight of cell material. A 2 mL aliquot of 18-megaohm deionized water was added to each vessel. Samples were spiked with 40 μL of 120 μg mL$^{-1}$ D6-lanosterol internal standard (Avanti Polar Lipids) in dichloromethane (DCM, Fisher Chemical). The solutions were sonicated by three 15-s, 6 W root mean squared (RMS) pulses while chilled on ice for cell lysis (Fisher Scientific 60 Sonic Dismembrator). The pH of each vessel was adjusted by the addition of 2 mL of 6% (w/w) potassium hydroxide in methanol. Vessels were sealed with crimp top caps (CEM Corp.), saponified at 90 °C for 1 h in an oven, and then cooled to room temperature (RT). A 2 mL aliquot of DCM was added through the septa, vessels were vortexed for 5 min and then centrifuged for 3 min to separate the aqueous and organic layers. Sterols were extracted with DCM two times by suction of the organic layer with a glass pipette and extracts were combined in a new sample vial (Supelco 27149). The DCM solvent was evaporated in vacuum at RT. Extracts were resuspended in 100 μL DCM and transferred to gas chromatography (GC) autosampler (AS) vials with 250 μL inserts (Restek). A second 100 μL DCM wash was used to rinse the sample vial and added to the GC AS vial for a more complete transfer. The DCM solvent was evaporated in vacuum at RT.

Derivatization of the dried extracts was performed by adding 40 μL of MSTFA (N-methyl-N-trimethylsilyltrifluoroacetamide; Sigma Aldrich) to the GC AS vial inserts, capping the vials, and heating at 40 °C for 30 min in an oven. The resulting solutions were directly analyzed by gas chromatography-mass spectrometry (GC-MS). A series of lanosterol standard solutions ranging from 50 to 450 μg mL$^{-1}$ were prepared, each with 120 μg mL$^{-1}$ D6-lanosterol internal standard and derivatized with MSTFA as described above.

**Gas chromatograph mass spectrometry and quantification**. Sterol quantification was performed using a gas chromatograph (Trace GC Ultra) equipped with an autosampler (Triplus AutoSampler, liquid injection) and interfaced with a single quadrupole mass spectrometer (DSQ II; all from Thermo Scientific). GC instrument settings were: splitless injection, 1 μL liquid injection volume, 220 °C injector temperature. The oven program was: 150 °C for 2 min, 20 °C min$^{-1}$ to 330 °C, hold 3 min. The transfer line temperature was held at 300 °C. Helium carrier gas was used at constant flow of 1.0 mL min$^{-1}$. The column used was a (5% phenyl)-95% dimethylsiloxane (Phenomenex ZB-5 30 m × 0.25 mm × 0.25 μm). MS settings were: positive ion mode with electron ionization, 70 eV electron energy, 200 μA filament emission current, 130 V lens, 250 °C source temperature, the quadrupole scanned *m/z* 50–650 with a scan time of 0.35 s. Samples, blanks, controls, and standards were randomized for injection order and injected in triplicate.

Lanosterol and parkeol quantification was performed with an internal standard (ISTD) multipoint calibration curve of lanosterol and D6-lanosterol as the ISTD. Peak area ratios of sterol (*m/z* 393) to ISTD (*m/z* 399) were measured in all samples and sterol content was computed from the calibration curve. Sterol content was then normalized to dry cell mass. For calculation of total sterol content, measurements for both sterols were summed. After testing for variance equivalency (*F*-test), the two-tailed Student's *t*-test at the 95% confidence interval was used to determine statistical support for differences in sterol content between different samples.

**Electron microscopy**. *G. obscuriglobus* mutant cells were scraped from the agar plates. The cells were frozen in an HPM010 (Abra Fluid, Switzerland) high-pressure freezing machine and freeze substituted with either 1% osmium tetroxide, 0.1% uranyl acetate, and 5% H$_2$O and embedded in Epon or with 0.5% uranyl acetate and embedded in Lowicryl HM20. Thin sections were placed on formvar-coated grids and post-stained with uranyl acetate and lead citrate. Thin sections were imaged on a CM120 Phillips electron microscope[25]. For terbinafine inhibition assays, *G. obscuriglobus* cultures were grown with or without terbinafine at 10, 20, or 50 μg mL$^{-1}$ for 4 days as described above. Cryofixation and freeze substitution were realized similarly and performed at the Molecular, Cellular, and Developmental Biology Electron Microscopy Facility, University of Colorado-Boulder. Ultrastructure images were acquired with use of a transmission electron microscope (Hitachi H-7000) at 75 kV, Gatan digital camera and Gatan software.

**Phase-contrast time-lapse microscopy**. *G. obscuriglobus* cells were grown for 4 days at 28 °C in PYGV liquid medium with 100 μg mL$^{-1}$ of terbinafine or without. 1 mL of each culture was pelleted, washed once in phosphate-buffered saline, and resuspended in sterile ddH$_2$O. This suspension was spotted onto a 1% agarose-PYGV pad on a glass slide. The pad contained 100 μg mL$^{-1}$ terbinafine for the treated cells. The pad was covered with a glass coverslip and sealed with a mixture of equal amount of petroleum jelly, lanolin, and paraffin wax (VALAP), leaving enough empty space around the agarose pad to provide the cells with sufficient oxygen. A single field of cells was imaged every 30 min during 14 h on a Zeiss Axio Imager Z2 epifluorescence microscope equipped with a Hamamatsu Orca-Flash4.0 sCMOS camera and a Plan-Apochromat 100×/1.46 Oil Ph3 objective. The images were collected in the Zen Blue software and processed using ImageJ and the EBImage package in Program R[26].

**Reporting summary**. Further information on research design is available in the Nature Research Reporting Summary linked to this article.

## Data availability

The authors declare that all other data supporting the findings of this study are available within the paper and its Supplementary Information files, or from the corresponding author on request. Source data are provided as a Source Data file.

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

## Acknowledgements
N.L.W. thanks Dr. Grant Bowman for use of microscopy equipment, and Dr. Ann Pearson for helpful discussions in the early stages of this project. E.R.-M., C.S.-M., and D. P.D. were supported by BFU2013-40866-P and BFU-2016-7826-P. S.S., E.Y.G., M.H., F. B., and N.L.W. were supported by the US National Science Foundation awards MCB-0920667 and IOS-1656637. S.S. and E.Y.G. were additionally supported by the US National Institutes of Health awards NCRR-P20RR016474 and NIGMS-P20GM103432. The acquisition of the GC-MS was made possible by the US National Science Foundation award CHE-0844694 to F.B.

## Author contributions
All authors designed the experiments. E.R.-M., S.S., E.Y.G., C.S.-M., and M.H. performed the experiments and analyzed the data. E.R.-M., S.S., E.Y.G., M.H., F.B., N.L.W., and D.P. D. wrote the manuscript. All authors reviewed and edited the manuscript. All authors read and approved the final manuscript.

## Additional information

**Competing interests:** The authors declare no competing interests.

