## [Peer Review File · Nature Communications]

Reviewers' comments:

Reviewer #1 (Remarks to the Author):

This manuscript shows indication that disruption of sterol synthesis has physiological implications in the bacterium *Gemmata obscuriglobus*. I have comments for the authors' consideration, in general, I found this work too preliminary. As the authors claim, the biological function and non-essential role of bacterial sterols is a fundamental point in biology (lines 44-45), yet this referee does not think this work addresses the point explicitly as the author claimed (lines 49-50) i.e. the question still stands. More experimental evidence is necessary to support the claims presented in this work.

1) the authors generate a sterol-depleted mutant in *Gemmata* and show that it is affected in growth. It grows poorly compared to the WT but this phenotype is however not "gene essentiality", as the mutant is able to grow and the authors were able to study the mutant using electron microscopy and biochemistry approaches. Genes that influence bacterial physiology are not essential genes.

2) The authors correlate lateral gene transfer (LGT) with a non-essential function of these genes. Then, they generate sterol-depleted mutants in *Gemmata* and, by showing that cell growth was affected in this mutant, suggested that sterol synthesis in *Gemmata* have not been acquired by LGT. This statement could be misleading, in such gene essentiality is directly related to growth conditions. Genes acquired by LGT can be essential to bacteria in conditions that are not necessarily those from laboratories. Vancomycin-resistance *Staphylococcus aureus* VRSA acquired the van operon from *Enterococcus* to change the cell wall structure and resist the vancomycin antimicrobial activity. The van operon is essential for *S. aureus* to grow in the presence of vancomycin. However, these cell wall structural changes generate a "low-quality" cell wall that makes VRSA to grow poorly in regular growth conditions in the absence of vancomycin.

- Also, regarding the phylogenetic analysis, it is unclear if the authors are interested in Oxidosqualene synthases as it says in line 52 or oxidosqualene cyclases as it says in figure legend 1.

- Line 55: "one group contained Oscs from Gammaproteobacteria and bacterioidetes." Is that OSC4 group? I see all options possible... OSC2 group contains only bacterioidetes and OSC1 contains only gamma proteobacteria. This is important because this rationale is what prompted the authors to use *Gemmata* as working model. I cannot see a special distribution of gammaproteobacteria and bacterioidetes in the tree.

3) Sterol mutant in *Gemmata* show aberrant cellular morphology, detected by electron microscopy. I find data in this section is very thin and important information is missing. Complementation of the mutants is necessary or comparing the effect to those treated with terbafine. The authors do not show the data ("data not shown" in line 106). By examining these phenotypes, several hypotheses are equally plausible, such as accumulation of toxic intermediates or even the toxicity of GFP in the mutants. The authors truncated the genes and inserted a *gfp* gene. They traced the stability of the mutants by following GFP expression. The GFP signal is however immunodetected in the extracellular space (see Figure 3g). It is possible that GFP expression is toxic and needs thus to be secreted, its toxicity generates cell aberrancies. With this, this referee wants just to illustrate that many possibilities can be supported with the experimental data that is in the manuscript. More data is necessary to clarify that these observations are due to the depletion of sterols in *Gemmata* and if so, whether these aberrant morphologies are causing poor growth in this bacterium.

As an additional comment, this manuscript contains many hand-waving statements; they are not necessary and many of them are overstatements. Some examples follow:

- Line 32, this is the first report of sterol essentiality in a prokaryotic species... This is not clear from the data that the authors show (see point 1 of this review).

- Line 80: "We speculate that..." This speculation needs to be removed because it has no support from the data
- Line 100: "Strikingly, phenotypes that have never been described in WT cells ..."
- Line 127: "the first reported evidence for an essential role of"
- Line 133: "To the best of our knowledge, this is also the first report of the..."

Reviewer #2 (Remarks to the Author):

The manuscript entitled "Essentiality of sterol genes in bacteria supports an independent evolutionary origin" focuses on subject that has been significantly understudied - sterol physiology in bacteria. As the authors point out, sterol synthesis and function has been studied extensively in eukaryotes and it is well-established that these lipids are critical to many cellular function in eukaryotic cells. Recent studies have begun to investigate the phylogeny and biosynthesis of these lipids in a broader group of bacterial species and the authors nicely summarize this previous work. But their study really begins to tease apart one of the most important unanswered questions - what are the functional roles of these lipids in bacterial cells. In addition, they enhance recent phylogenetic studies that have begun to hint at an independent origin of sterol synthesis in some bacteria species. It has been assumed that the few known bacteria sterol producers acquired sterols synthesis genes through horizontal gene transfer from an ancient eukaryote. But this study, which combines a phylogenetic and physiological experimental approaches, provides further evidence that the evolutionary history of sterol synthesis is more complicated and does support that in some lineages this biosynthetic pathway has a bacterial origin. Overall, I think this is an excellent study and a well-written manuscript. I find the essentially of sterols and a potential role for these lipids in cell division in Gemmata very exciting. This opens up a whole new area of study in sterol biology that definitely deserves to be pursued.

Below are some specific points that I think need to be addressed:

1. Line 42-43 - The authors here make a claim that is disseminated widely in the literature - that hopanoids and sterols are chemically, structurally, and functionally analogous. The one study they cite focuses on the membrane ordering effects of hopanoids in vesicles which showed a similar, albeit reduced, effect to what is observed with sterols. My main issue with that study is that the hopanoid used (diploptene) is not the functional form of hopanoids found in cells and therefore, that "analogous" role is, in my opinion, not supported. Also, chemically and structurally, sterols and hopanoids are not analogous. Sterols have four core rings and hopanoids have five; sterols are derived from oxidosqualene and hopanoids from squalene; sterols have a hydrophilic hydroxyl group at the A-ring and hopanoids do not; hopanoids have a very hydrophilic tail with multiple hydroxyl groups (and in some cases amine groups) and sterols have a very hydrophobic tail. These differences are significant and are going to affect how sterols and hopanoids interact with membranes (and potentially proteins). Given how the authors are showing a very important role for sterols in the viability of Gemmata, we need to begin to emphasize that these lipids are different and may play different roles in bacterial cells.

2. Line 44-45 - My understanding is that sterol synthesis has only been shown to be indispensable through genetic analysis in one myxobacterial strain, *Stigmatella*. I am not sure if there are other studies that have shown they are indispensable in other bacteria through other methods (such as terbinafine induced depletion). So I think this statement can be more accurate and say that this has been shown in one organism. Also ref 14 is the appropriate one to cite for this statement (which the authors do) but ref 4 is not. That study only looked at the production of sterols in a variety of bacterial species and did not assess whether sterols were dispensable in any of the bacteria tested.

3. Line 56-59 - In the discussion of the phylogeny of bacterial OSC, the authors point out that one group (OSC4) has a clear LGT origin. Based on the topology, I think this is a correct statement. But they go on to say that this is "consistent with the non-essential nature of sterols in representative Proteobacterial species from this group." That statement is inaccurate and should be removed. As mentioned in my second point above, the only bacterium in which sterol synthesis has been genetically deleted is *Stigmatella*, a Delta Proteobacterium, which I am guessing is clustering in OSC3 not in OSC4. It would also help to have an uncollapsed version of the Figure 1 tree in the supplement so we could see where the different organisms are falling (the outgroup can remain collapsed so that the tree is able to fit on one page). Alternatively, a supplemental table listing the organisms in the different OSC groups could be included.

4. Line 88-91 - In describing the terbinafine experiments, the authors do not state what the mechanism of inhibition might be. My recollection is that terbinafine inhibits *Sqmo*, which would result in the accumulation of squalene in their cultures. In addition, blocking *Sqmo* genetically should also result in a significant accumulation of squalene. Do the authors observe squalene accumulation in their lipid analyses? Is it possible that any phenotypes observed are a result of squalene accumulation rather than sterol depletion?

5. Line 117 - Could the phase bright inclusions be excessive squalene?

6. Line 140-145 - The use of sterols as molecular markers in paleogeological studies is actually much more nuanced than what the authors are stating here. Most sterol biomarkers are diagenetic products of cholesterol, ergosterol, stigmaterol - eukaryotic sterols that are much more enzymatically modified than the sterols produced by Gemmata and other bacteria. In particular, the eukaryotic sterol biomarkers do not have methylations at the C-4 position and have alkylations at the C-24 position that are actually diagnostic. So the occurrence of lanosterol and parkeol in Gemmata and the potential ancestry they put forth in this paper does not, in my opinion, confound the current interpretation of steranes in the fossil record.

7. Line 165 - Ah, my soapbox issue! LB stands for lysogeny broth not Luria-Bertani. Giuseppe Bertani clarified this himself in the *Journal of Bacteriology* in 2004 (Bertani G. (2004) Lysogeny at Mid-Twentieth Century: P1, P2, and Other Experimental Systems *J. Bacteriol.* 186: 595-600. doi: 10.1128/JB.186.3.595-600.2004).

Paula Welander
Stanford University

Reviewer #3 (Remarks to the Author):

In this work, Rivas-Marin et al. have conducted a number of phylogenetic and functional analyses that show that the bacterium *Gemmata obscuriglobus*, one of the few bacteria that synthesize sterols (usually considered a marker of eukaryotes):

- Has not acquired the genes required for sterol synthesis by lateral gene transfer from eukaryotes.
- Elimination of sterols (by genetic or chemical methods) from *G. obscuriglobus* considerably impairs its growth.

I have the following major issues with this work:

1. The authors observe slow growth in the absence of sterols. They "speculate that the sterol-deleted

colonies were able to grow, although inefficiently, by dilution of residual membrane sterols". This is a speculation and I do not think that their findings demonstrate essentiality, which is one of the main points of this article, mentioned in the title.

2. Even if essentiality of sterols was demonstrated in this group, I am not sure how important/impactful this finding would be. The fact that something is essential does not demonstrate an independent origin.

3. They mention that "In the sqmo mutant, we observed diverse striking phenotypes that have never been described in wild-type cells". The fact that something has not been described does not mean that it does not exist, and if one looks for abnormalities, it is a matter of time before one finds them. The authors should perform a systematic comparison for wild-type vs. mutated cells, rather than just reporting abnormalities. I do not doubt, however, that there will be differences, I just think that they should be reported in a more systematic way.

4. The title seems inappropriate to me. It is not only essentiality, but also phylogenetic analyses, that support an independent evolutionary origin. In addition, independent evolutionary origin in what group? (in *G. obscuriglobus* I presume).

5. In any case, even if these concerns were addressed, I would not think that this work is of sufficient interest for Nature Communications. I think that a revised version of this manuscript would be suitable for a more specialized journal.

Reviewers' comments:

Reviewer #1 (Remarks to the Author):

This manuscript shows indication that disruption of sterol synthesis has physiological implications in the bacterium *Gemmata obscuriglobus*. I have comments for the authors' consideration, in general, I found this work too preliminary. As the authors claim, the biological function and non-essential role of bacterial sterols is a fundamental point in biology (lines 44-45), yet this referee does not think this work addresses the point explicitly as the author claimed (lines 49-50) i.e. the question still stands. More experimental evidence is necessary to support the claims presented in this work.

Reply: We agree that the work described here is only a first attempt at addressing the issue. Since submission, we have performed additional experiments that further support our claim. Together, we believe that the data is now interesting enough to be disseminated to the community. In addition, we have considerably turned down the evolutionary implications of the results.

1) the authors generate a sterol-depleted mutant in *Gemmata* and show that it is affected in growth. It grows poorly compared to the WT but this phenotype is however not "gene essentiality", as the mutant is able to grow and the authors were able to study the mutant using electron microscopy and biochemistry approaches. Genes that influence bacterial physiology are not essential genes.

R: We are sorry to disagree, but we do not think that the mutant is able to grow. As indicated, only a couple of divisions appears to be possible. This mutant is unable to grow when tiny colonies are re-inoculated on solid or liquid medium, no growth at all is observed. The most likely explanation for this very limited growth is the dilution of the sterol originally present in the mother cell membranes. In addition, frequency of trans-conjugants is much smaller than when a non-essential gene is disrupted. Taken together, we do believe that the data supports essentiality of the targeted genes.

Due to the lack of growth of the insertion mutant, electron microscopy directly on these tiny colonies is one of the few experiment that we are able to perform. However, the biochemical approach is carried out using wild type cultures grown in conditions (terbinafine) in which the enzyme *Sqmo* is chemically inhibited. This inhibition is performed

at low drug concentration to only partially inhibit the enzyme. Higher concentrations impede growth, as expected for an essential gene.

2) The authors correlate lateral gene transfer (LGT) with a non-essential function of these genes. Then, they generate sterol-depleted mutants in *Gemmata* and, by showing that cell growth was affected in this mutant, suggested that sterol synthesis in *Gemmata* have not been acquired by LGT. This statement could be misleading, in such gene essentiality is directly related to growth conditions. Genes acquired by LGT can be essential to bacteria in conditions that are not necessarily those from laboratories. Vancomycin-resistance *Staphylococcus aureus* VRSA acquired the van operon from *Enterococcus* to change the cell wall structure and resist the vancomycin antimicrobial activity. The van operon is essential for *S. aureus* to grow in the presence of vancomycin. However, these cell wall structural changes generate a "low-quality" cell wall that makes VRSA to grow poorly in regular growth conditions in the absence of vancomycin.

R: We believe that this example is not appropriate as it describe the action of a damaging agent, while our work addresses a compound required for growth. This is in fact the exact opposite. Indeed, it is well known that antibiotic resistance, by example, can be transferred by LGT, but this is not the same as a necessary metabolite.

- Also, regarding the phylogenetic analysis, it is unclear if the authors are interested in Oxidosqualene synthases as it says in line 52 or oxidosqualene cyclases as it says in figure legend 1.

- Line 55: "one group contained Oscs from Gammaproteobacteria and bacteroidetes." Is that OSC4 group? I see all options possible... OSC2 group contains only bacteroidetes and OSC1 contains only gamma proteobacteria. This is important because this rationale is what prompted the authors to use Gemmata as working model. I cannot see a special distribution of gammaproteobacteria and bacteroidetes in the tree.

R: Both comments have now been corrected or clarified.

3) Sterol mutant in Gemmata show aberrant cellular morphology, detected by electron microscopy. I find data in this section is very thin and important information is missing. Complementation of the mutants is necessary or comparing the effect to those treated with terbafine. The authors do not show the data ("data not shown" in line 106).

R: Complementation of the mutant providing lanosterol in trans in the medium was attempted, unfortunately to no effect. As there are no replicative plasmid described for Planctomycetes, the closest to complementation assay is provided by the experiment where we insert a plasmid that is similar but that reconstruct the whole gene. In this case, no phenotype is observed demonstrating that the observed phenotypes are due to the sterol pathway interruption. In addition, we do now provide comparison with terbinafine-treated colonies. Although, defects are milder (due to low drug concentration), they are similar.

By examining these phenotypes, several hypotheses are equally plausible, such as accumulation of toxic intermediates or even the toxicity of GFP in the mutants. The authors truncated the genes and inserted a gfp gene. They traced the stability of the mutants by following GFP expression. The GFP signal is however immunodetected in the extracellular space (see Figure 3g). It is possible that GFP expression is toxic and needs thus to be secreted, its toxicity generates cell aberrancies. With this, this referee wants just to illustrate that many possibilities can be supported with the experimental data that is in the manuscript. More data is necessary to clarify that these observations are due to the depletion of sterols in Gemmata and if so, whether these aberrant morphologies are causing poor growth in this bacterium.

R: It would be very surprising to realize that GFP is toxic to the cell. Indeed, literally hundreds of GFP expressing mutants have been described in the literature, a very limited amount of which (unknown to us) might report GFP toxicity. In addition, our 'complementation' control also expresses GFP under the same promoter and no toxicity effect has been observed despite the fact that the cells are green. We believe that altogether, our data is supporting our claim that the phenotypes observed are due to sterol

depletion, and not to an unlikely GFP toxicity effect. Again, additional experiments have been performed to support our claims.

Concerning the fact that GFP signal is detected in the extracellular space, this is clearly due to the fact that some of the cells are broken and intracellular material is released.

As an additional comment, this manuscript contains many hand-waving statements; they are not necessary and many of them are overstatements. Some examples follow:

- Line 32, this is the first report of sterol essentiality in a prokaryotic species... This is not clear from the data that the authors show (see point 1 of this review).

- Line 80: "We speculate that..." This speculation needs to be removed because it has no support from the data

- Line 100: "Strikingly, phenotypes that have never been described in WT cells ..."

- Line 127: "the first reported evidence for an essential role of"

- Line 133: "To the best of our knowledge, this is also the first report of the..."

R: Those have been removed or tuned-down in the current version.

Reviewer #2 (Remarks to the Author):

The manuscript entitled "Essentiality of sterol genes in bacteria supports an independent evolutionary origin" focuses on subject that has been significantly understudied - sterol physiology in bacteria. As the authors point out, sterol synthesis and function has been studied extensively in eukaryotes and it is well-established that these lipids are critical to many cellular functions in eukaryotic cells. Recent studies have begun to investigate the phylogeny and biosynthesis of these lipids in a broader group of bacterial species and the authors nicely summarize this previous work. But their study really begins to tease apart one of the most important unanswered questions - what are the functional roles of these lipids in bacterial cells. In addition, they enhance recent phylogenetic studies that have begun to hint at an independent origin of sterol synthesis in some bacteria species. It has been assumed that the few known bacterial sterol producers acquired sterol synthesis genes through horizontal gene transfer from an ancient eukaryote. But this study, which combines a phylogenetic and physiological experimental approaches, provides further evidence that the evolutionary history of sterol synthesis is more complicated and does support that in some lineages this biosynthetic pathway has a bacterial origin. Overall, I think this is an excellent study and a well-written manuscript. I find the essentiality of sterols and a potential role for these lipids in cell division in Gemmata very exciting. This opens up a whole new area of study in sterol biology that definitely deserves to be pursued.

Below are some specific points that I think need to be addressed:

1. Line 42-43 - The authors here make a claim that is disseminated widely in the literature - that hopanoids and sterols are chemically, structurally, and functionally analogous. The one study they cite focuses on the membrane ordering effects of hopanoids in vesicles which showed a similar, albeit reduced, effect to what is observed with sterols. My main issue with that study is that the hopanoid used (diploptene) is not the functional form of hopanoids found in cells and therefore, that "analogous" role is, in my opinion, not supported. Also, chemically and structurally, sterols and hopanoids are not analogous. Sterols have four core rings and hopanoids have five; sterols are derived from oxidosqualene and hopanoids from squalene; sterols have a hydrophilic hydroxyl group at the A-ring and hopanoids do not; hopanoids have a very hydrophilic tail with multiple hydroxyl groups (and in some cases amine groups) and sterols have a very hydrophobic tail. These differences are significant and are going to affect how sterols and hopanoids interact with membranes (and potentially proteins). Given how the authors are showing a very important role for sterols in the viability of Gemmata, we need to begin to emphasize that these lipids are different and may play different roles in bacterial cells.

R: We thank the reviewer for this comment. We have indeed modified the text accordingly. Note that we meant 'analogue' for compounds in the sense of 'chemically similar'. The term was probably not correct. We have modified it and highlighted the differences.

2. Line 44-45 - My understanding is that sterol synthesis has only been shown to be indispensable through genetic analysis in one myxobacterial strain, *Stigmatella*. I am not sure if there are other studies that have shown they are indispensable in other bacteria through other methods (such as terbinafine induced depletion). So I think this statement can be more accurate and say that this has been shown in one organism. Also ref 14 is the appropriate one to cite for this statement (which the authors do) but ref 4 is not.

That study only looked at the production of sterols in a variety of bacterial species and did not assess whether sterols were dispensable in any of the bacteria tested.

R: Corrected.

3. Line 56-59 - In the discussion of the phylogeny of bacterial OSC, the authors point out that one group (OSC4) has a clear LGT origin. Based on the topology, I think this is a correct statement. But they go on to say that this is "consistent with the non-essential nature of sterols in representative Proteobacterial species from this group." That statement is inaccurate and should be removed. As mentioned in my second point above, the only bacterium in which sterol synthesis has been genetically deleted is *Stigmatella*, a Delta Proteobacterium, which I am guessing is clustering in OSC3 not in OSC4. It would also help to have an uncollapsed version of the Figure 1 tree in the supplement so we could see where the different organisms are falling (the outgroup can remain collapsed so that the tree is able to fit on one page). Alternatively, a supplemental table listing the organisms in the different OSC groups could be included.

R: We have corrected the statement and now proved an uncollapsed version of the tree as supplementary figure.

4. Line 88-91 - In describing the terbinafine experiments, the authors do not state what the mechanism of inhibition might be. My recollection is that terbinafine inhibits *Sqmo*, which would result in the accumulation of squalene in their cultures. In addition, blocking *Sqmo* genetically should also result in a significant accumulation of squalene. Do the authors observe squalene accumulation in their lipid analyses? Is it possible that any phenotypes observed are a result of squalene accumulation rather than sterol depletion?

R: The mechanism of action of terbinafine is now described. In addition, we provide new experimental data addressing the point of squalene accumulation by treatment with zaragozic acid.

5. Line 117 - Could the phase bright inclusions be excessive squalene?

R: This is indeed very likely, now commented in the text.

6. Line 140-145 - The use of sterols as molecular markers in paleogeological studies is actually much more nuanced than what the authors are stating here. Most sterol biomarkers are diagenetic products of cholesterol, ergosterol, stigmasterol - eukaryotic sterols that are much more enzymatically modified than the sterols produced by Gemmata and other bacteria. In particular, the eukaryotic sterol biomarkers do not have methylations at the C-4 position and have alkylations at the C-24 position that are actually diagnostic. So the occurrence of lanosterol and parkeol in Gemmata and the potential ancestry they put forth in this paper does not, in my opinion, confound the current interpretation of steranes in the fossil record.

R: ok, modified in the text.

7. Line 165 - Ah, my soapbox issue! LB stands for lysogeny broth not Luria-Bertani. Giuseppe Bertani clarified this himself in the *Journal of Bacteriology* in 2004 (Bertani G. (2004) Lysogeny at Mid-Twentieth Century: P1, P2, and Other Experimental Systems *J. Bacteriol.* 186: 595-600. doi: 10.1128/JB.186.3.595-600.2004).

R: Corrected.

Paula Welander
Stanford University

Reviewer #3 (Remarks to the Author):

In this work, Rivas-Marín et al. have conducted a number of phylogenetic and functional analyses that show that the bacterium *Gemmata obscuriglobus*, one of the few bacteria that synthesize sterols (usually considered a marker of eukaryotes):

- Has not acquired the genes required for sterol synthesis by lateral gene transfer from eukaryotes.

- Elimination of sterols (by genetic or chemical methods) from *G. obscuriglobus* considerably impairs its growth.

I have the following major issues with this work:

1. The authors observe slow growth in the absence of sterols. They “speculate that the sterol-deleted colonies were able to grow, although inefficiently, by dilution of residual membrane sterols”. This is a speculation and I do not think that their findings demonstrate essentiality, which is one of the main points of this article, mentioned in the title.

R: We have now removed that point from the title.

2. Even if essentiality of sterols was demonstrated in this group, I am not sure how important/impactful this finding would be. The fact that something is essential does not demonstrate an independent origin.

R: We have attempted to minimize the speculation about origin in the current version. Note however that the novelty is not on the independent origin but on the essentiality of sterol in bacteria which is the first such report for prokaryotes (see reviewer #2's comments).

3. They mention that “In the *sqmo* mutant, we observed diverse striking phenotypes that have never been described in wild-type cells”. The fact that something has not been described does not mean that it does not exist, and if one looks for abnormalities, it is a matter of time before one finds them. The authors should perform a systematic comparison for wild-type vs. mutated cells, rather than just reporting abnormalities. I do not doubt, however, that there will be differences, I just think that they should be reported in a more systematic way.

R: We now provide this comparison with WT cells. Please note that electron-microscopy has for a long time been the only way to study Planctomycetes. Years of study have looked at Planctomycetes cell membranes. So despite the statement of the reviewer, we do believe that the observed phenotypes are indeed due to our interruption of the genes instead of a question of time. We believe that you could look at Gemmata WT for a very long time and not observe the phenotypes we describe here. However, the scarcity and slow growth of the mutant combined to the long delay before getting enough material and enough pictures to look at render a more systematic analysis of the EM data quite difficult.

4. The title seems inappropriate to me. It is not only essentiality, but also phylogenetic analyses, that support an independent evolutionary origin. In addition, independent evolutionary origin in what group? (in *G. obscuriglobus* I presume).

R: modified.

5. In any case, even if these concerns were addressed, I would not think that this work is of sufficient interest for Nature Communications. I think that a revised version of this manuscript would be suitable for a more specialized journal.

Reviewers' comments:

Reviewer #1 (Remarks to the Author):

The manuscript Essentiality of sterol synthesis genes in the planctomycete bacterium *Gemmata obscuriglobus* is a revised version of a previous submission. In the previous version, this referee commented on several technical aspects of the manuscript that overall make this study too preliminary for publication in Nature Communications and made some experimental recommendations to the authors, to improve the quality of the work and generate more conclusive experimental data. However, the authors did not address any of these experimental questions in this revised version or in the response letter to the referees' comments. In the response letter, the authors disagree on or reject any of the technical possibilities proposed to solve the major technical issues of this manuscript while accepting all textual and minor modifications. The manuscript is very similar to the previous version. This referee does not think this current version of the manuscript has improved during this revision process. The manuscript is as thin in data as it was the previous version and data is too preliminary, drawing conclusions that sometimes are not supported by the data that is presented. As this referee pointed in the previous revision, more experimental evidence is necessary to support the claims presented in this work.

Comments follow:

The sterol mutant generates small colonies which impedes further propagation in liquid or solid medium. The essentiality of the gene was a matter of discussion in the previous version, as the mutant is able to grow and generate small colonies. As referred in the previous version, the mutant grows poorly compared to the WT but this does not support gene essentiality. The authors claimed that this phenotype is probably consequence of the dilution of the sterol coming from the mother cell membranes but this is an speculation. In toto, these phenotype needs to be supported with more robust data.

In the previous version, the authors correlate lateral gene transfer (LGT) with the non-essential function of the genes. They thus considered that sterol synthesis genes, since they are essential for the bacterium, were not acquired by LGT. Again, this is a bold statement that needs to be supported by robust piece of data. In the previous version, this referee just pointed an example of how this is not always true. The authors claimed that the example is not valid but they do not modify any aspect of this claim.

These are the three major issues that still stand:

- As stated in the previous comment, the essentiality of the sterol synthesis genes still needs to be demonstrated. It is difficult in these conditions to establish any correlation between gene essentiality and lateral gene transfer.
- Analysing living organisms in laboratory conditions does not allow determining whether the natural conditions that favour lateral gene transfer make the transferring gene essential for the life of the bacterium. The example of *S. aureus* is clear but discussing the suitability of examples is futile. How can the authors prove that natural conditions in which lateral gene transfer occur generated a non-essential environment for the expression of these gene, just by growing the bacterium in laboratory conditions?
- the authors need to prove better that this is the case, that non-essentiality applies to genes mobilized by lateral gene transfer and that this applies to the sterol synthesis genes in *Gemmata*.

From the previous letter; Referee: I cannot see a special distribution of gammaproteobacteria and bacteroidetes in the tree.

Authors: Both comments have now been corrected or clarified.

In this revised version, this referee still cannot see a special distribution of gammaproteobacteria and bacteriodetes in the tree.

The electron microscopy data showing aberrant cellular morphology in the sterol mutant in Gemmata is very thin. More experiments are needed to draw any conclusion from this section. In the response letter, the authors claimed that they performed experiments in which they generated a complemented mutant and detected that this mutant showed no cellular morphologies defect. This referee cannot find this data in the revised version of the paper. In contrast, the experiment with terbafine, which produces milder effect in cellular morphology and does not support the conclusions drawn in this section, is shown in supplemental figure 5. Overall, this section is very confusing and needs further experimentation.

The authors truncated the genes and inserted a gfp gene. They traced the stability of the mutants by following GFP expression. The GFP signal is however immunodetected in the extracellular space. Again, experiments are few and thus conclusions are not sufficiently supported. Many conclusions apply to this result. In the previous letter, this referee pointed out a simple example to illustrate that more than one conclusion can be drawn from this result. The authors again argued about the suitability of the example. As stated before, the suitability of the example is a minor issue, the authors should focus on providing enough experimental data to clearly support their conclusions but in this particular case I will follow the argument.

The authors should read more about GFP expression and toxicity because it is relatively usual that the expression of a GFP protein affects negatively cellular metabolism. The authors used the classical GFP variant, which tends to interact and form aggregates that are toxic for the bacterium. GFP can aggregate with and titrate other bacterial proteins thus reducing considerably cellular operations. Additionally, it is relatively usual to observe the accumulation of GFP aggregates in the cellular membrane, causing severe damage to the cell, which in turn generates cell disruption. Because all this, many laboratories have dedicated efforts in the past twenty years to develop a number of GFP variants with reduce toxic effects. For instance, there are mGFP variants (monomeric GFP) that do not aggregate, as well as venus variant... But the use of these variants do not preclude the presence of toxic effects.

In summary, the fact GFP signal is detected in the extracellular space is indicative that some cells are dying and they liberate cell debris. However, with the experiments that are presented in this paper, it is not possible to determine yet whether cell lysis is consequence of sterol depletion or the effect of GFP expression in the cells, as it sometimes occurs in many other different systems.

Reviewer #2 (Remarks to the Author):

This revised manuscript is much improved over the original submission. The authors have adequately addressed my comments and their added experiments with the zaragozic acid treatment further supports their main conclusions. I think the difficulty of working with Gemmata is clear and I commend them for a very nice study in a difficult microbial system.

Paula Welander
Stanford University

Reviewers' comments:

Reviewer #1 (Remarks to the Author):

The manuscript Essentiality of sterol synthesis genes in the planctomycete bacterium *Gemmata obscuriglobus* is a revised version of a previous submission. In the previous version, this referee commented on several technical aspects of the manuscript that overall make this study too preliminary for publication in Nature Communications and made some experimental recommendations to the authors, to improve the quality of the work and generate more conclusive experimental data. However, the authors did not address any of these experimental questions in this revised version or in the response letter to the referees' comments. In the response letter, the authors disagree on or reject any of the technical possibilities proposed to solve the major technical issues of this manuscript while accepting all textual and minor modifications. The manuscript is very similar to the previous version. This referee does not think this current version of the manuscript has improved during this revision process. The manuscript is as thin in data as it was the previous version and data is too preliminary, drawing conclusions that sometimes are not supported by the data that is presented. As this referee pointed in the previous revision, more experimental evidence is necessary to support the claims presented in this work.

Reply: As pointed out in our reply but also by reviewer #2, experimental work with this non-model organism is limited. However, we believe that we have been extremely cautious at the time of interpreting our results and drawing conclusions. We also believe that we have taken the necessary controls to sustain our claims.

Comments follow:

The sterol mutant generates small colonies which impedes further propagation in liquid or solid medium. The essentiality of the gene was a matter of discussion in the previous version, as the mutant is able to grow and generate small colonies. As referred in the previous version, the mutant grows poorly compared to the WT but this does not support gene essentiality. The authors claimed that this phenotype is probably consequence of the dilution of the sterol coming from the mother cell membranes but this is an speculation. In toto, these phenotype needs to be supported with more robust data.

Reply : As stated previously, it is not true that the mutant grows poorly, the mutant is unable to display unlimited growth it is only able to sustain a limited number of cell divisions. This is most likely the results of the 'dilution' of the sterol present in the cell as well as the one synthesized by the enzyme until exhaustion. The point is however not on why the mutant can divide a limited amount of time, the point is that it can not do so indefinitely. The statement that the mutant grows poorly compared to the WT is simply untrue. The mutant does not grow.

In the previous version, the authors correlate lateral gene transfer (LGT) with the non-essential function of the genes. They thus considered that sterol synthesis genes, since they are essential for the bacterium, were not acquired by LGT. Again, this is a bold statement that needs to be supported by robust piece of data. In the previous version, this referee just pointed an example of how this is not always true. The authors claimed that the example is not valid but they do not modify any aspect of this claim.

Reply: Please note that the LGT aspect has been considerably turned down in the revised version and we only mention it, as should be, but do not make this a major point of the manuscript, not even mentioning it in the discussion.

These are the three major issues that still stand:

- As stated in the previous comment, the essentiality of the sterol synthesis genes still needs to be demonstrated. It is difficult in these conditions to establish any correlation between gene essentiality and lateral gene transfer.

Reply: This point has been addressed previously. Again, please take into account the fact that the LGT issue is not a major claim of the manuscript.

- Analysing living organisms in laboratory conditions does not allow determining whether the natural conditions that favour lateral gene transfer make the transferring gene essential for the life of the bacterium. The example of *S. aureus* is clear but discussing the suitability of examples is futile. How can the authors prove that natural conditions in which lateral gene transfer occur generated a non-essential environment for the expression of these genes, just by growing the bacterium in laboratory conditions?

- the authors need to prove better that this is the case, that non-essentiality applies to genes mobilized by lateral gene transfer and that this applies to the sterol synthesis genes in Gemmata.

Reply: Again, this is an incorrect reading of our manuscript, as the phylogeny analysis does not agree with an origin from lateral gene transfer. Why do we need to prove better that genes mobilized by lgt are unessential while the subject of our study is unlikely to be the result of lgt?

From the previous letter; Referee: I cannot see a special distribution of gammaproteobacteria and bacterioidetes in the tree.

Authors: Both comments have now been corrected or clarified.

In this revised version, this referee still cannot see a special distribution of gammaproteobacteria and bacterioidetes in the tree.

Reply: It is unclear to us why the reviewer cannot see a special distribution of these two groups. They are the only ones forming the group OSC4, the only group to have a clear origin from LGT. According to the suggestions of both reviewers we modified the manuscript to move this description of the phylogenetic tree at the end of the manuscript.

The electron microscopy data showing aberrant cellular morphology in the sterol mutant in Gemmata is very thin. More experiments are needed to draw any conclusion from this section. In the response letter, the authors claimed that they performed experiments in which they generated a complemented mutant and detected that this mutant showed no cellular morphology defect. This referee cannot find this data in the revised version of the paper. In contrast, the experiment with terbafine, which produces milder effect in cellular morphology and does not support the conclusions drawn in this section, is shown in supplemental figure 5. Overall, this section is very confusing and needs further experimentation.

Reply: It is unclear to us why the reviewer thinks that the terbafine experiment does not support the conclusions of the section. This is however one of the novel experiments addressing the previous comments of the reviewers.

The authors truncated the genes and inserted a gfp gene. They traced the stability of the mutants by following GFP expression. The GFP signal is however immunodetected in the extracellular space. Again, experiments are few and thus conclusions are not sufficiently supported. Many conclusions apply to this result. In the previous letter, this referee pointed out a simple example to illustrate that more than one conclusion can be drawn from this result. The authors again argued about the suitability of the example. As stated before, the suitability of the example is a minor issue, the authors should focus on providing enough experimental data to clearly support their conclusions but in this particular case I will follow the argument.

The authors should read more about GFP expression and toxicity because it is relatively usual that the expression of a GFP protein affects negatively cellular metabolism. The authors used the classical GFP variant, which tends to interact and form aggregates that are toxic for the bacterium. GFP can aggregate with and titrate other bacterial proteins thus reducing considerably cellular operations. Additionally, it is relatively

usual to observe the accumulation of GFP aggregates in the cellular membrane, causing severe damage to the cell, which in turn generates cell disruption. Because all this, many laboratories have dedicated efforts in the past twenty years to develop a number of GFP variants with reduce toxic effects. For instance, there are mGFP variants (monomeric GFP) that do not aggregate, as well as venus variant... But the use of these variants do not preclude the presence of toxic effects.

In summary, the fact GFP signal is detected in the extracellular space is indicative that some cells are dying and they liberate cell debris. However, with the experiments that are presented in this paper, it is not possible to determine yet whether cell lysis is consequence of sterol depletion or the effect of GFP expression in the cells, as it sometimes occurs in many other different systems.

Reply: As pointed out in the introduction, the reviewer just ignores the fact that our complemented mutant also contain GFP and do not present any of the phenotypes supposedly associated with GFP toxicity. The cells are indeed dying, but this is due to the lack of sterol, as the complemented mutant also contain GFP and do not show this phenotype.

Reviewer #2 (Remarks to the Author):

This revised manuscript is much improved over the original submission. The authors have adequately addressed my comments and their added experiments with the zaragozic acid treatment further supports their main conclusions. I think the difficulty of working with Gemmata is clear and I commend them for a very nice study in a difficult microbial system.

Paula Welander
Stanford University